# Integrated RNA-seq Analysis Indicates Asynchrony in Clock Genes between Tissues under Spaceflight

**DOI:** 10.3390/life10090196

**Published:** 2020-09-11

**Authors:** Shin-ichiro Fujita, Lindsay Rutter, Quang Ong, Masafumi Muratani

**Affiliations:** 1Doctoral Program in Biomedical Sciences, Graduate School of Comprehensive Human Sciences, University of Tsukuba, Ibaraki 305-8575, Japan; s1730411@s.tsukuba.ac.jp (S.-i.F.); s1736043@s.tsukuba.ac.jp (Q.O.); 2Department of Genome Biology, Faculty of Medicine, University of Tsukuba, Ibaraki 305-8575, Japan; rutter.lindsay.fn@u.tsukuba.ac.jp

**Keywords:** circadian rhythm, RNA-seq, bioinformatics, genomics, gene expression, microgravity, spaceflight, space biology

## Abstract

Rodent models have been widely used as analogs for estimating spaceflight-relevant molecular mechanisms in human tissues. NASA GeneLab provides access to numerous spaceflight omics datasets that can potentially generate novel insights and hypotheses about fundamental space biology when analyzed in new and integrated fashions. Here, we performed a pilot study to elucidate space biological mechanisms across tissues by reanalyzing mouse RNA-sequencing spaceflight data archived on NASA GeneLab. Our results showed that clock gene expressions in spaceflight mice were altered compared with those in ground control mice. Furthermore, the results suggested that spaceflight promotes asynchrony of clock gene expressions between peripheral tissues. Abnormal circadian rhythms are associated not only with jet lag and sleep disorders but also with cancer, lifestyle-related diseases, and mental disorders. Overall, our findings highlight the importance of elucidating the causes of circadian rhythm disruptions using the unique approach of space biology research to one day potentially develop countermeasures that benefit humans on Earth and in space.

## 1. Introduction

The unique environment during spaceflight affects various physiological functions in astronauts [1,2]. With increasing opportunities for human spaceflight, a better understanding of the effects of spaceflight on the human body at the molecular level is essential. Unfortunately, solid tissue biopsies cannot be performed in the full human body in most cases. Human biopsies can usually only be obtained by using less invasive approaches, rendering it all the more difficult to investigate human tissue-level mechanisms during spaceflight. The impact of spaceflight itself is challenging to understand owing to its inherent complexity of environmental stressors, such as confinement in a closed environment, microgravity, radiation, and noise. Since the dawn of human spaceflight, basic life science experiments have been actively conducted in space to learn more about disease processes here on Earth from a new and unique perspective. Rodent models are often used as analogs for estimating biological mechanisms inside human tissues [3,4]. Mice studies under spaceflight conditions have been successfully conducted to elicit molecular mechanisms at the tissue and cellular levels [3,5,6,7,8].

Omics analysis has led to many novel discoveries for researchers through advances in genome-wide analysis techniques [9]. Since the NASA GeneLab database was launched in 2015, the open-access database and its multi-omics format have provided researchers with countless new insights [10,11,12,13,14,15]. The NASA GeneLab database stores a variety of open-access, spaceflight-related omics datasets for numerous organisms, including microbes, plants, animals, and humans [16,17]. Datasets are stored as microarrays, RNA-seq, bisulfite sequencing, proteomics, and metabolomics [16,17]. Omics data are rich with information on the different molecules that make up living organisms. Furthermore, GeneLab Analysis Working Groups (AWGs) regularly update data standardization workflows in the database to meet the latest recommendations from the bioinformatics discipline. As such, multi-omics data from a platform like NASA GeneLab can be continuously re-investigated with newer and deeper analyses to potentially extract novel insights and hypotheses that assess the biological risks of space missions.

This study aimed to reanalyze archived RNA-seq data from NASA GeneLab to unravel molecular responses to spaceflight. We conducted enrichment analysis on data from multiple tissues to investigate what molecular changes were revealed under spaceflight conditions. Enrichment analysis indicated that common ontologies and critical regulators between peripheral tissues were associated with circadian rhythms. Our results suggested that clock genes across tissues were affected by expression changes under spaceflight conditions. Furthermore, we observed asynchrony in the expression of the clock genes between certain peripheral tissues. Circadian rhythm disruption is associated not only with astronaut health and performance but also with cancer, lifestyle-related diseases, and mental disorders here on Earth [18,19]. Therefore, further studies on how spaceflight affects tissue functions should be an essential agenda item in order to better understand mammalian physiology and potentially create countermeasures that benefit humans on Earth and in space. Our study did not determine whether clock genes were primarily affected by spaceflight factors (such as microgravity and radiation) and/or extraneous factors (such as rearing environment and sample processing) [20]. Our findings not only generate new insights into space biological mechanisms, but also underline the critical need for space biology researchers to maintain and share detailed experimental metadata so that meta-analyses may one day disentangle exactly which factors cause circadian rhythm disruption during spaceflight.

## 2. Materials and Methods

### 2.1. Data Preparation and Processing

The transcriptome datasets used for this study were from openly available data housed on NASA’s GeneLab platform and are updated to meet the latest recommendations from the analysis workflow by NASA GeneLab AWGs (genelab.nasa.gov). Specifically, 8 RNA-seq datasets from NASA GeneLab (GLDS-98, -99, -101, -102, -103, -104, -105, and -168) were used for downstream analysis [21,22,23,24,25,26,27,28]. These data were all derived from the same mission. Metadata on NASA GeneLab indicated that 16-week-old C57BL/6 J female mice were used and that the mice were maintained under a 12 h light/dark cycle throughout the 37-day spaceflight mission. Each dataset corresponded to a different mouse tissue: adrenal glands, extensor digitorum longus muscle, gastrocnemius muscle, kidneys, quadriceps muscle, soleus muscle, tibialis anterior muscle, and liver, respectively. In addition, each dataset contained about 5–6 spaceflight (FLT) and 5–6 ground control (GC). We used the External RNA Control Consortium (ERCC) results for the quantitative values of GLDS-168. We performed 2 main forms of differentially expressed gene (DEG) analyses: 1 tissue-wide analysis and 8 individual tissue analyses. For the tissue-wide analysis, we compared all the FLT samples regardless of the tissue (*n* = 47) with all GC samples regardless of the tissue (*n* = 46). Quantile normalization was performed on each “rna_seq_Unnormalized counts.csv” file stored in each “GeneLab Processed RNA-Seq Files” directory on NASA GeneLab using CLC Genomics Workbench (version 11.0.2, QIAGEN, Hilden, Germany). The *t* test was used for the statistical analysis, and the statistical threshold was defined with a false discovery rate (FDR) < 0.1. For each of the 8 individual tissue analyses, we downloaded the “rna_seq_differential_expression.csv” files stored in the “GeneLab Processed RNA-Sseq Files” directory on NASA GeneLab, in which statistical analysis had already been conducted to determine DEGs. The statistical threshold was defined as FDR < 0.05.

### 2.2. Data Visualization

Principal component analysis (PCA) was performed using the scikit-learn package 0.17.1 in Python v3.7 (https://scikit-learn.org). Heatmaps were generated using the Python seaborn.clustermap function (https://seaborn.pydata.org).

### 2.3. Enrichment Analysis

Enrichment analysis was performed on DEGs using Metascape with default settings [29].

## 3. Results

### 3.1. Spaceflight Could Cause Significant Changes in the Expression of Clock Genes in Multiple Tissues

To examine how spaceflight affects prevalent molecular mechanisms across tissues, we reanalyzed 8 RNA-seq datasets archived on the NASA GeneLab database using tissue-wide analysis. Comparison between FLT and GC identified 13 up-regulated genes and 13 down-regulated genes (FDR < 0.1) (Figure 1A). PCA plots of the 26 tissue-wide DEGs showed that the first principal component separated the 5 muscle tissues from the adrenal glands, kidneys, and liver tissues, whereas the second principal component separated the FLT group from the GC group (Figure 1B). Enrichment analysis showed that terms related to circadian rhythms, herpes simplex infection, and endocrine system development were significantly enriched in the 26 DEGs of the tissue-wide analysis (Figure 1C).

### 3.2. Spaceflight Could Enrich Key Regulators Related to Circadian Rhythms in Peripheral Tissues

DEGs between FLT and GC were identified at the FDR < 0.05 level separately for each of the 8 individual tissues: adrenal glands (*n* = 67), kidneys (*n* = 299), liver (*n* = 4644), extensor digitorum longus (*n* = 2809), gastrocnemius (*n* = 232), quadriceps (*n* = 763), soleus (*n* = 4931), and tibialis anterior (*n* = 1163) (Figure 2A). A previous report that analyzed these data [13] found that spaceflight induced the largest number of DEGs in the soleus (similar to our current results). However, despite this consistent ranking, our results were not consistent with the previous study in terms of the actual DEG counts [13]. We note that Beheshti et al. used the same dataset as we did but analyzed it using a different RNA-seq workflow that included a t-test with a *p*-value ≤ 0.05 [13]. This indicates that RNA-seq results highly depend on the standardized processing workflows. To allow sufficient reproducibility, the NASA Genelab team has established standardized processing workflows for omics datasets (https://genelab-data.ndc.nasa.gov/genelab/projects). Our study used DEGs that were defined by analysis pipelines developed by GeneLab AWGs, which regularly update and share data standardization workflows in the database to meet the latest recommendations from the bioinformatics discipline. Therefore, the reanalysis of archived datasets might be worthwhile for finding novel insights. Our findings showed that the liver was the tissue with the second largest number of DEGs. This discrepancy may be due to the liver datasets being re-sequenced between the previous report and our study as described in GLDS-48 (https://genelab-data.ndc.nasa.gov/genelab/accession/GLDS-48). PCA plots demonstrated remarkable differences between FLT and GC conditions in all tissues (Figure 2B). These results suggested that gene expression changes occurred in each peripheral tissue during spaceflight. To examine the underlying molecular mechanisms across tissues caused by spaceflight, enrichment analysis was performed on the DEGs of the 8 tissues and showed shared terms between tissues (Figure 3A). Clustering analysis of protein–protein interaction (PPI) networks using MODE showed that the terms related to circadian rhythms were most significantly enriched in MODE1 (Figure 3B,C).

### 3.3. Spaceflight Could Induce Asynchrony in Clock Genes between Peripheral Tissues

The genes in MODE1 are known to play a role in a core loop that regulates circadian rhythm cycles via a feedback mechanism, altogether synchronizing circadian rhythms between tissues from signals of the central nervous system via light input to the eye (Figure 3B) [30]. Therefore, we predicted that clock genes under spaceflight would show gene expression patterns that were consistent between peripheral tissues. To investigate this hypothesis, we focused on the genes in the MODE1 cluster and confirmed their FDR values (Figure 4A). We found that some clock genes did not significantly change expression during spaceflight in the adrenal glands, kidneys, and liver (Figure 4A). For example, *Cry* and *Per* gene expressions were not significantly altered under spaceflight in the adrenal glands (Figure 4A). On the other hand, most muscle tissues showed a significantly changed expression for most of these same clock genes under spaceflight (Figure 4A). Furthermore, we plotted the fold changes of *Arntl* (also known as *Bmal1*) and *Per2* as representative clock genes because they are known to be in antiphase with each other (Figure 4B) [31]. *Arntl* was consistently up-regulated and *Per2* was consistently down-regulated across all muscle tissues in the spaceflight condition (Figure 4B). *Arntl* was also consistently up-regulated in the adrenal glands, kidneys, and liver. However, *Per2* did not show significant changes in the adrenal glands and liver, and only showed significant changes at less conservative thresholds in the kidneys compared to the muscle tissues (Figure 4B). In summary, our results suggested that the clock genes became asynchronous between certain peripheral tissues by some undetermined stimuli under spaceflight conditions.

## 4. Discussion

Our purpose was to elucidate the biological mechanisms across tissues by reanalyzing integrated RNA-seq data of mice during spaceflight. We found that the FLT group showed significantly altered clock gene expressions across tissues compared to the GC group (Figure 1A–C). Astronauts experience sleep and circadian rhythm shifts that impair health, alertness, and performance during spaceflight [18,19]. Therefore, understanding the effects of the spaceflight environment on the circadian rhythm system is important for long-term crewed space missions. It would also be important as the commercial spaceflight field expands for passengers and pilots who regularly fly orbital or suborbital flights between, say, New York City and Tokyo. This is because the circadian rhythms of the cells in most peripheral tissues are uncoupled into their own circadian rhythms by crossing time zones in a jet plane or spaceflight [32]. Moreover, enrichment analysis in the tissue-wide analysis enriched herpes simplex infection and endocrine system development as prevalent molecular mechanisms across tissues (Figure 1C). Previous studies have reported immune alterations during spaceflight, consisting of reductions in T cell and Natural Killer (NK) cell function, and the reactivation of latent herpes viruses [33,34,35]. Spaceflight could induce increased levels of stress hormones including cortisol, dehydroepiandrosterone, epinephrine, and norepinephrine through hypothalamic-pituitary-adrenal (HPA) and sympathetic-adrenal-medullary (SAM) axes activation [33,34,35]. These stress hormones are reported to affect the immune system such as causing the reactivation of viruses under spaceflight [33,34,35]. Circulating stress hormones, such as glucocorticoids, play a role in the mediation of the circadian clock and the synchronization of peripheral clocks [36,37]. Therefore, the interaction of the endocrine axis and the immune system with regard to the circadian clock could be associated with the underlying molecular mechanisms across tissues under spaceflight.

The central clock localized in the suprachiasmatic nucleus (SCN) unifies circadian rhythms between tissues through neurotransmission and hormonal signals [30]. After the SCN receives light input from the eye, the resulting signal from the SCN synchronizes the peripheral clocks of peripheral tissues. At the same time, this light-induced mechanism can be altered by external stimuli such as food intake, which can cause peripheral tissues to perform their own phase of circadian rhythm [31]. In other words, other events may be driving the observed circadian rhythm changes other than the light–dark cycle in peripheral tissues. Indeed, previous mice studies during spaceflight have implemented carefully controlled 12 h light–dark cycles but still observed circadian rhythm-related disruptions at both the behavioral and molecular biological levels. For instance, studies have reported that the FLT group showed significant increases in behavior activity during the dark cycle and in food intake throughout the mission compared to the GC group [3,38]. Likewise, other studies reported that clock-related gene expressions were changed in the liver and eyes of mice that underwent spaceflight for about one month [10,30,39]. A previous study suggested that the circadian rhythm could contribute to signs of non-alcoholic fatty liver disease (NAFLD) at the molecular level in mice under spaceflight [10]. It is essential for future studies to investigate the molecular details of how circadian rhythms are disrupted during spaceflight.

Our results using DEGs from the 8 tissues showed that circadian rhythms were commonly enriched (Figure 3A). Clustering analysis of PPI networks showed that MODE1 was most significantly enriched in terms and genes related to the circadian rhythm (Figure 3B,C). Genes in MODE1 positively regulate clock gene expressions, such as those of *Per* and *Cry*, by forming heterodimers of the transcription factors (TFs) Arntl and Clock [31]. On the other hand, the proteins produced from these same clock genes negatively regulate the transcriptional activities of *Arntl* and *Clock* to form the core loop of the diurnal fluctuation rhythm [31]. We found that most of the clock genes in the MODE1 cluster showed significantly altered expression across the muscle tissues under spaceflight (Figure 4A). In contrast, most clock genes (except *Arntl*) in the MODE1 cluster did not significantly change in the adrenal glands, kidneys, and liver (Figure 4A). While the fold change of *Arntl* was consistently up-regulated across all tissues, *Per2* did not show significant changes in the adrenal glands and liver, and only showed significant changes at less conservative thresholds in the kidneys compared to the muscle tissues (Figure 4B). Given that the data we used came from studies that reported treating GC mice as similarly as possible to FLT mice by implementing controlled light–dark cycles and food resources within the same housing devices [3,38], we predicted that clock genes under spaceflight would show gene expression patterns that were in synchrony between peripheral tissues. However, our results suggested that other factors besides the environmental light–dark cycle (such as gravity, radiation, and rearing environment) may be driving asynchrony of the circadian rhythm between certain peripheral tissues during spaceflight.

We observed that the adrenal glands, kidneys, and liver showed similar expression distribution in the PCA plot (Figure 1B). Enrichment analysis for the tissue-wide analysis showed that endocrine system development terms were enriched as prevalent molecular mechanisms across tissues. In addition, these 3 tissues showed inconsistent clock gene expression changes compared to most muscle tissues (Figure 4A,B). These 3 tissues relate to functions known to be affected by the circadian rhythm, such as the endocrine systems (renin-angiotensin-aldosterone system, erythropoietin, vasopressin, and glucocorticoids) and water–mineral balance regulation [40,41,42,43]. Indeed, spaceflight conditions in humans are believed to induce activation of the endocrine systems, including renin-aldosterone, glucocorticoid, and catecholamines, and changes in water–mineral balance [18,44,45,46]. In addition, during a Mars simulation study in humans, the hormones aldosterone and cortisol fluctuated for longer-than-usual periods despite constant salt intake, suggesting that clock genes may be involved in water–mineral balance [47]. Few studies have examined the responses of adrenal glands and kidneys in spaceflight [45,47,48,49,50,51]. Therefore, our observed asynchrony of clock genes in these three tissues may provide new molecular-level insights about the endocrine systems and water–mineral balance regulation during spaceflight.

*Arntl*, *Clock*, *Per2*, *Per3*, *Cry1*, and *Cry2* were enriched across some tissues in MODE1 of the PPI network analysis (Figure 3B). In addition, *Arntl*, *Clock*, and *Per3* were significantly changed in the tissue-wide analysis (Figure 1A). Since TFs of these genes form core clock components in circadian rhythm mechanisms [31], these clock genes could play a role as key regulators associated with underlying mechanisms of physiology and metabolism under spaceflight [18,52]. For example, previous studies showed that clock genes have demonstrated a role of circadian rhythm in muscle atrophy and bone remodeling [53,54,55]. In addition, these clock genes have been implicated in NAFLD [56], the symptoms of which have been observed in mice during spaceflight [10]. Furthermore, *Arntl* plays a role in the circadian regulation of acute glucocorticoid secretion in the adrenal glands in response to stress [57]. Mice models carrying a conditional allele for *Arntl* showed several disorders, such as increased urine volume, changes in the circadian rhythm of urinary sodium excretion, increased glomerular filtration rate, and significantly reduced plasma aldosterone levels [58]. *Clock* null mice showed abnormal circadian rhythmicity of plasma aldosterone levels and changes in circadian gene expression patterns in the kidney [59]. Glucocorticoids could shift the phase of circadian oscillations of *Per1* and *Per2* expressions in peripheral tissues [60]. *Cry* genes are associated with changes in the transcriptional response to glucocorticoids in mouse embryonic fibroblasts [61]. In addition, *Cry1* and *Cry2* null mice indicated salt-sensitive hypertension via abnormally high synthesis of the mineralocorticoid aldosterone by the adrenal gland [62]. Given that circulating stress hormones are increased and circadian rhythms are altered in astronauts and mice [34,35,63], spaceflight could contribute to circadian rhythm disruption and asynchrony between peripheral tissues. Therefore, these results suggest that these clock genes could modulate physiology and metabolism mechanisms as key regulators during spaceflight environmental changes. However, the role of clock genes is unknown in peripheral tissues under spaceflight. In a simulated microgravity study, for example, *Arntl* disrupted diurnal oscillation in rat cerebrovascular contractility by changing circadian regulation of the miR-103/CaV 1.2 signal pathway [64]. Oscillations of *Arntl* were amplified under simulated microgravity in human keratinocytes [65]. Future studies could determine clock gene roles in peripheral tissues under spaceflight, such as by using clock genes null mice under spaceflight [66].

This study has several limitations. First, the RNA-seq data in this study only reflect the effects of spaceflight after 1-month missions. We note that major clock gene expressions of *Drosophila melanogaster* were unaffected after short-term spaceflight for 13 days, despite being exposed to the same 12 h light/dark cycles that the mice were exposed to in the data we analyzed [20]. Hence, our pilot study provides new motivation to elucidate both shorter and longer time series of molecular circadian mechanisms under spaceflight in mammals. Second, it is unclear whether our mice results generalize to human levels; future spaceflight experiments can explore clock-related gene expression changes in human tissue samples derived from liquid biopsies. Third, although the data we used came from studies that reported treating GC mice as similarly as possible to FLT mice, clock gene expressions may be changed by factors that may not be routinely controlled within and between spaceflight datasets, such as sample collection and dissection schedules [20,31]. It is crucial for principal investigators to provide detailed metadata when publishing their data; this includes sample handling information, such as dissection and freezing schedules. In addition, the current study provided preliminary findings related to circadian rhythm disruptions during spaceflight using limited types of tissue datasets that were all from the same mission. However, the circadian rhythm is known to depend on other systems that we did not explore (such as cardiovascular and nervous systems) [18]; these types of data are currently not available on NASA GeneLab collectively from the same mission and sample manipulation. Future studies may expand upon our pilot study and investigate all systems related to circadian rhythms if they become available in NASA GeneLab.

In summary, since the safety and performance of astronauts and commercial spaceflight pilots can be affected by disruption of circadian clocks, our study highlights the necessity of further investigation into the impact of spaceflight on mammalian tissue functions. Our results shed novel insights into possible health consequences under spaceflight conditions, including the overall disruption of clock gene synchronization between peripheral tissues. As of now, our pilot study remains inconclusive as to whether disruptions in the clock gene expressions in mice during spaceflight are due to the space environment itself or to sample operation artifacts or to both. The current cross-data study was possible due to open-source space omics databases that implement state-of-the-art metadata normalization. As metadata standardization and multi-omics approaches to space biology continue to improve, future studies may more conclusively determine causes and countermeasures to circadian rhythm disruptions in space.

## Figures and Tables

**Figure 1 life-10-00196-f001:**
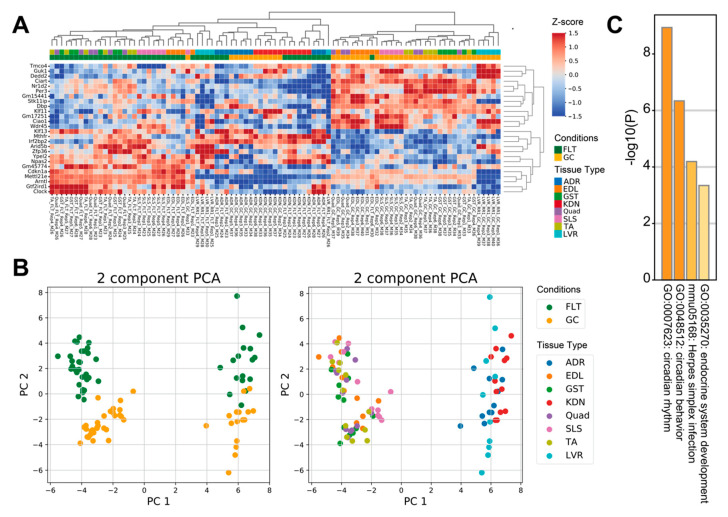
Differential gene expression between spaceflight and ground control across mice tissues. (**A**) is the heatmap of the 26 tissue-wide DEGs between FLT and GC. (**B**) are PCA plots of the 26 tissue-wide DEGs between FLT and GC conditions. PCA plots are colored by conditions (left) and tissue types (right). (**C**) is the bar chart of enrichment ontology categories in the tissue-wide DEGs between FLT and GC. Abbreviations: FLT, spaceflight mice; GC, ground control mice; ADR, adrenal glands; EDL, extensor digitorum longus; GST, gastrocnemius; KDN, kidneys; Quad, quadriceps; SLS, soleus; TA, tibialis anterior; LVR, liver.

**Figure 2 life-10-00196-f002:**
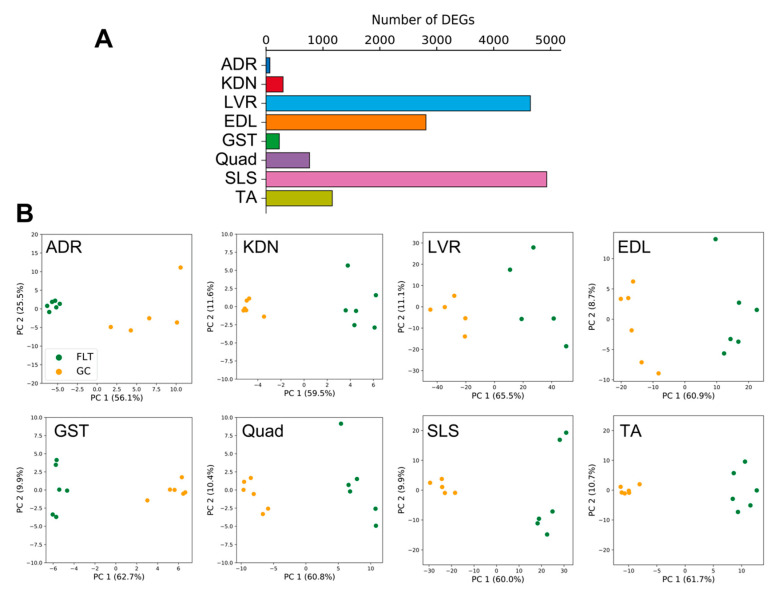
DEG numbers and PCA plots for each peripheral tissue. (**A**) is the number of DEGs detected between FLT and GC conditions. (**B**) are PCA plots for each of the 8 different tissues.

**Figure 3 life-10-00196-f003:**
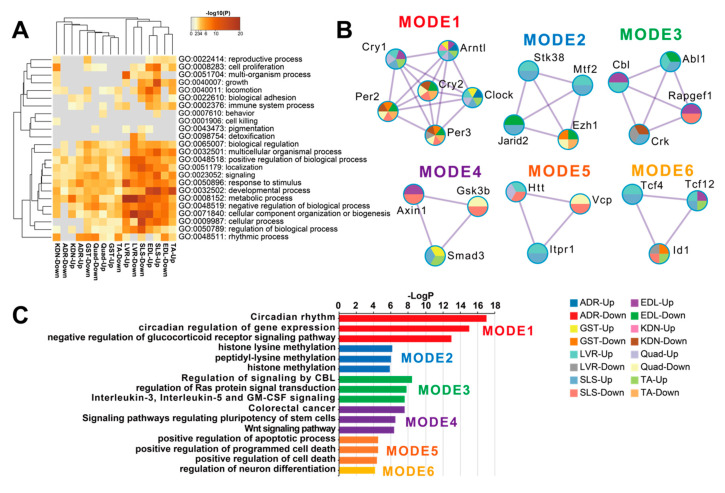
Functional enrichment analysis of DEGs in each tissue. (**A**) is the heatmap of the top-level Gene Ontology (GO) biological process categories for DEGs in each tissue. Gray color indicates a lack of significant term. (**B**) are the clustered PPI networks by MCODE identified from the combined list of all DEGs in each tissue. (**C**) is the bar plot of clusters of enrichment categories detected by the MCODE algorithm. Abbreviations: Up, up-regulated genes; Down, down-regulated genes.

**Figure 4 life-10-00196-f004:**
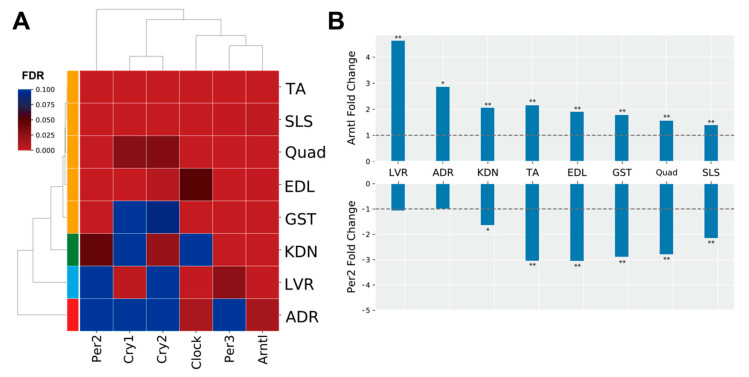
Clock gene expression patterns. (**A**) is the heatmap of the FDR values of clock genes for each tissue. (**B**) is the fold change (FLT/GC) of representative clock genes (*Arntl* and *Per2*), which are known to be in antiphase oscillation with each other. The dashed horizontal lines indicate fold change magnitudes with values of one as visual references. * FDR < 0.05, ** FDR < 0.01.

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
