# Peer review of "Integrated RNA-seq Analysis Indicates Asynchrony in Clock Genes between Tissues under Spaceflight"

_life, 2020, doi:10.3390/life10090196_

Round 1
Reviewer 1 Report
The pilot study assess the impact of spaceflight on tissue collected from rodent mission flown to the ISS. The analyses are based on RNA-seq data available from the NASA GeneLab database. The comparative analyses revealed difference in gene expression and more notably related to clock genes. Overall the manuscript is well written and the data are well presented. General comments would be related to some suggested references and maybe more insight from the discussion.
Specific Comments:
Line 28: I would recommend the newly released book on principles of clinical medicine for spaceflight (DOI 10.1007/978-1-4939-9889-0) as an additional reference.
Line 41: Suggestion, here is a good review on omics technologies for space applications as an additional reference (Biotechnol Adv. 2017 Nov 15;35(7):905-932. doi: 10.1016/j.biotechadv.2017.04.003. Epub 2017 Apr 19.)
Line 44-45: Add the following Ref ( https://pubmed.ncbi.nlm.nih.gov/30329036/)
Line 73-74: Please double check Life's instruction if website for the database should be included in the reference 18-25.
Line 78: define DEG (see line 86)
Line 90: Reference?
Line 116: Can you please address commonality and difference between your study and other (Ref 12) for soleus either here or in the discussion.
Line 118: Any evidence that the data changed? I think GeneLab is good at keeping a log of the changes.
Line 179 and 193: Can you please reconcile these two opposite statements related to liver gene expression? Maybe in the discussion section?
Line 213: Add reference
Line 221: Add reference
Figure 1A: Why the muscle tissue specific replicates don't cluster well together as opposed to other tissue? Interestingly GC seems to cluster better. Any comments? Maybe related to tissue preservation?
Figure 1C: The authors didn't comment herpes simplex infection finding from the enrichment ontology. Any specific insights? Many studies on crew health have shown herpes virus reactivation while in space (https://pubmed.ncbi.nlm.nih.gov/32464118/)
Fig 2A: Please keep the same nomenclature as in Fig 1B. Tissue classified by alphabetic order.
Author Response
Response to Reviewer 1 Comments
Point 1: Line 28: I would recommend the newly released book on principles of clinical medicine for spaceflight (DOI 10.1007/978-1-4939-9889-0) as an additional reference.
Response: Thank you for your comment. We have added the reference according to your suggestion.
Point 2 Line 41: Suggestion, here is a good review on omics technologies for space applications as an additional reference (Biotechnol Adv. 2017 Nov 15;35(7):905-932. doi: 10.1016/j.biotechadv.2017.04.003. Epub 2017 Apr 19.)
Response: As per your suggestion, we have added the reference.
Point 3 Line 44-45: Add the following Ref (https://pubmed.ncbi.nlm.nih.gov/30329036/)
Response: As you suggested, we have added the reference.
Point 4 Line 73-74: Please double check Life's instruction if website for the database should be included in the reference 18-25.
Response: Thank you for this advice. We found instructions on Life’s website about including DOIs in references to previously-published data. As a result, we added the references for each of the 8 GeneLab datasets (with their DOIs) to the Reference section. To be extra careful, we are also leaving a note with the editor to double-check that this is the intended format for referencing datasets in Life.
Point 5 Line 78: define DEG (see line 86)
Response: According to your suggestion, we modified the sentence below.
“For each of the 8 individual tissue analyses, we downloaded the "rna_seq_differential_ expression.csv" files stored in the “GeneLab Processed RNA-Sseq Files” directory on NASA GeneLab, in which statistical analysis had already been conducted to determine DEGs."
Point 6 Line 90: Reference?
Response: In line with your suggestion, we added the reference.
Point 7 Line 116: Can you please address commonality and difference between your study and other (Ref 12) for soleus either here or in the discussion.
Response: Thank you for your comment. According to your suggestion, we modified the sentence below.
“A previous report that analyzed these data [13] found that spaceflight induced the largest number of DEGs in the soleus (similar to our current results). However, despite this consistent ranking, our results were not consistent with the previous study in terms of the actual DEG counts [13]. We note that Beheshti et al. used the same dataset as we did but analyzed it using a different RNA-seq workflow that included a t-test with a p-value ≤ 0.05 [13]. This indicates that RNA-seq results highly depend on the standardized processing workflows. To allow sufficient reproducibility, the NASA Genelab team has established standardized processing workflows for omics datasets (https://genelab-data.ndc.nasa.gov/genelab/projects). Our study used DEGs that were defined by analysis pipelines developed by GeneLab AWGs, which regularly update and share data standardization workflows in the database to meet the latest recommendations from the bioinformatics discipline. Therefore, the reanalysis of archived datasets might be worthwhile for finding novel insights.”
Point 8 Line 118: Any evidence that the data changed? I think GeneLab is good at keeping a log of the changes.
Response: The NASA Gene Lab has the following statement on GLDS-48 (https://genelab-data.ndc.nasa.gov/genelab/accession/GLDS-48).
"RNA extracted from these tissues was re-sequenced; these data are available as part of GLDS-168 (https://genelab-data.ndc.nasa.gov/genelab/accession/GLDS-168)."
According to your suggestion, we modified the sentence below.
“Our findings showed that the liver was the tissue with the second largest number of DEGs. This discrepancy may be due to the liver datasets being re-sequenced between the previous report and our study as described in GLDS-48 (https://genelab-data.ndc.nasa.gov/genelab/accession/GLDS-48).”
Point 9 Line 179 and 193: Can you please reconcile these two opposite statements related to liver gene expression? Maybe in the discussion section?
Response: According to your suggestion, we revised the sentence below.
“In contrast, most clock genes (except Arntl) in the MODE1 cluster did not significantly change in the adrenal glands, kidneys, and liver (Figure 4A).”
Point 10 Line 213: Add reference
Response: We took into account your suggestion and added the reference.
Point 11 Line 221: Add reference
Response: In line with your recommendation, we added the reference and additional discussion about other genes below.
“Arntl, Clock, Per2, Per3, Cry1, and Cry2 were enriched across some tissues in MODE1 of the PPI network analysis (Figure 3B). In addition, Arntl, Clock, and Per3 were significantly changed in the tissue-wide analysis (Figure 1A). Since TFs of these genes form core clock components in circadian rhythm mechanisms [31], these clock genes could play a role as key regulators associated with underlying mechanisms of physiology and metabolism under spaceflight [18,52]. For example, previous studies showed that clock genes have demonstrated a role of circadian rhythm in muscle atrophy and bone remodeling [53–55]. In addition, these clock genes have been implicated in NAFLD [56], the symptoms of which have been observed in mice during spaceflight [10]. Furthermore, Arntl plays a role in the circadian regulation of acute glucocorticoid secretion in the adrenal glands in response to stress [57]. Mice models carrying a conditional allele for Arntl showed several disorders, such as increased urine volume, changes in the circadian rhythm of urinary sodium excretion, increased glomerular filtration rate, and significantly reduced plasma aldosterone levels [58]. Clock null mice showed abnormal circadian rhythmicity of plasma aldosterone levels and changes in circadian gene expression patterns in the kidney [59]. Glucocorticoids could shift the phase of circadian oscillations of Per1 and Per2 expressions in peripheral tissues [60]. Cry genes are associated with changes in the transcriptional response to glucocorticoids in mouse embryonic fibroblasts [61]. In addition, Cry1 and Cry2 null mice indicated salt-sensitive hypertension via abnormally high synthesis of the mineralocorticoid aldosterone by the adrenal gland [62]. Given that circulating stress hormones are increased and circadian rhythms are altered in astronauts and mice [34,35,63], spaceflight could contribute to circadian rhythm disruption and asynchrony between peripheral tissues. Therefore, these results suggest that these clock genes could modulate physiology and metabolism mechanisms as key regulators during spaceflight environmental changes. However, the role of clock genes is unknown in peripheral tissues under spaceflight. In a simulated microgravity study, for example, Arntl disrupted diurnal oscillation in rat cerebrovascular contractility by changing circadian regulation of the miR-103/CaV 1.2 signal pathway [64]. Oscillations of Arntl were amplified under simulated microgravity in human keratinocytes [65]. Future studies could determine clock gene roles in peripheral tissues under spaceflight, such as by using clock genes null mice under spaceflight [66].”
Point 12 Figure 1A: Why the muscle tissue specific replicates don't cluster well together as opposed to other tissue? Interestingly GC seems to cluster better. Any comments? Maybe related to tissue preservation?
Response: Thank you for your comment. According to your suggestion, we also considered tissue preservation for this clustering result. However, it was unclear to us. Muscle tissues did not cluster well together by hierarchical clustering partly because different muscle types have similar expression profiles to each other compared to other tissues. Indeed, PCA plots of DEGs showed that the muscle tissues are clustered together (Figure 1B). Therefore, it can be inferred that hierarchical clustering does not accumulate by tissue specific replicates.
Point 13 Figure 1C: The authors didn't comment herpes simplex infection finding from the enrichment ontology. Any specific insights? Many studies on crew health have shown herpes virus reactivation while in space (https://pubmed.ncbi.nlm.nih.gov/32464118/)
Response: Thank you for your comment. Regarding your comment, we added the following discussion.
“This is because the circadian rhythms of the cells in most peripheral tissues are uncoupled into their own circadian rhythms by crossing time zones in a jet plane or spaceflight [32]. Moreover, enrichment analysis in the tissue-wide analysis enriched herpes simplex infection and endocrine system development as prevalent molecular mechanisms across tissues (Figure 1C). Previous studies have reported immune alterations during spaceflight, consisting of reductions in T cell and Natural Killer (NK) cell function, and the reactivation of latent herpes viruses [33–35]. Spaceflight could induce increased levels of stress hormones including cortisol, dehydroepiandrosterone, epinephrine, and norepinephrine through hypothalamic-pituitary-adrenal (HPA) and sympathetic-adrenal-medullary (SAM) axes activation [33–35]. These stress hormones are reported to affect the immune system such as causing the reactivation of viruses under spaceflight [33–35]. Circulating stress hormones, such as glucocorticoids, play a role in the mediation of the circadian clock and the synchronization of peripheral clocks [36,37]. Therefore, the interaction of the endocrine axis and the immune system with regard to the circadian clock could be associated with the underlying molecular mechanisms across tissues under spaceflight.”
Point 14 Fig 2A: Please keep the same nomenclature as in Fig 1B. Tissue classified by alphabetic order.
Response: As you suggested, we modified the figure.

Reviewer 2 Report
Abnormal circadian rhythms are associated with several pathologies. In their manuscript “Integrated RNA-seq analysis indicates asynchrony in clock genes between tissues under spaceflight” Shin-ichiro Fujita and coauthors report about clock gene expression alterations in mice after a stay in NASA’s GeneLab onboard the ISS. Especially clock gene expressions between peripheral tissues showed an asynchrony which may explain possible health consequences for humans under spaceflight conditions. It should be mentioned that this was an in silico study and authors reanalyzed archived RNA-seq data from NASA GeneLab (doi:10.3791/58447). Nevertheless, the analysis was well performed, data is well presented, the manuscript is accurately written, and the overall claim is supported by the results. I appreciate that the authors also listed the limitations of the study in the discussion. Nevertheless, I have the following suggestions for the current version of the manuscript, which may require the authors’ further consideration.
- I don't understand why the study was limited to these 8 tissues (adrenal glands, extensor digitorum longus, muscle, gastrocnemius muscle, kidneys, quadriceps muscle, soleus muscle, tibialis anterior muscle, liver) when using existing data. There are 4 “major” systems affected by the circadian clock: musculoskeletal system, nervous system, cardiovascular system and - very important - the endocrine system that should be analyzed to draw any general conclusions about tissues.
- M&M: Is there any information about the mice (type, age, …?) and the conditions (light–dark cycle, etc.? - here I would suggest doi:10.1126/science.11540800 or some Drosophila experiments for comparison) in GeneLab?
- In Figure 1A you showed a gene expression signature which is likely the basis of this analysis. With exception of Arntl, none of these genes are described in the results section and / or in the discussion. However, other genes are found there (Cry, Per2, ….). Why this discrepancy?
Author Response
Response to Reviewer 2 Comments
Point 1: I don't understand why the study was limited to these 8 tissues (adrenal glands, extensor digitorum longus, muscle, gastrocnemius muscle, kidneys, quadriceps muscle, soleus muscle, tibialis anterior muscle, liver) when using existing data. There are 4 “major” systems affected by the circadian clock: musculoskeletal system, nervous system, cardiovascular system and - very important - the endocrine system that should be analyzed to draw any general conclusions about tissues.

Response 1: Thank you for your suggestion. Unfortunately, NASA GeneLab does not store all tissue datasets experienced during spaceflight. However, the eight datasets that we analyzed were all from the same spaceflight mission and underwent the same sample manipulation, allowing for particular control of potential confounding variables introduced between missions. This was one of the larger available collections of datasets that were made by the same spaceflight mission and underwent the same sample manipulation. Therefore, we selected this particular set of datasets for analysis in order to optimize the sample number and minimize confounding variables between them.
As you suggested, we added information in the discussion as shown below.
“In addition, the current study provided preliminary findings related to circadian rhythm disruptions during spaceflight using limited types of tissue datasets that were all from the same mission. However, the circadian rhythm is known to depend on other systems that we did not explore (such as cardiovascular and nervous systems) [18]; these types of data are currently not available on NASA GeneLab collectively from the same mission and sample manipulation. Future studies may expand upon our pilot study and investigate all systems related to circadian rhythm if they become available in NASA GeneLab.”
Point 2: M&M: Is there any information about the mice (type, age, …?) and the conditions (light–dark cycle, etc.? - here I would suggest doi:10.1126/science.11540800 or some Drosophila experiments for comparison) in GeneLab?
Response 2: Thank you for your suggestion. As you suggested, we added information in M&M as shown below.
“These data were all derived from the same mission. Metadata on NASA GeneLab indicated that 16-week-old C57BL/6 J female mice were used and that the mice were maintained under a 12-hour light/dark cycle throughout the 37-day spaceflight mission.”
We also added more information about Drosophila melanogaster spaceflight studies that used similar 12-hour light/dark cycles but importantly did not observe expression changes in major circadian rhythm genes. We modified this in a sentence in the Discussion section as follows:
“We note that major clock gene expressions of Drosophila melanogaster were unaffected after short-term spaceflight for 13 days, despite being exposed to the same 12-hour light/dark cycles that the mice were exposed to in the data we analyzed [20]”
A previous study in 2015 also treated Drosophila melanogaster to 12-hour light/dark cycles and noted that major clock gene expressions were unaffected after short-term spaceflight for 13 days [1].
Even though mice have been treated to carefully-controlled 12-hour light-dark cycles during spaceflight, we observed clock gene expression changes in peripheral tissues. Therefore, other events may be driving the observed circadian rhythm changes other than the light-dark cycle in peripheral tissues. The RNA-seq data in this study only reflect the effects of spaceflight after 1-month missions. Hence, our pilot study provides new motivation to elucidate both shorter and longer time series of molecular circadian mechanisms under spaceflight in mammals.
However, our study did not determine whether clock genes were primarily affected by spaceflight factors (such as microgravity and radiation) and/or extraneous factors (such as rearing environment and sample processing) [1]. This is an important point because a previous spaceflight study discussed that clock gene expression in Drosophila melanogaster was affected by sample manipulation [1]. Our findings also underlined the critical need for space biology researchers to maintain and share detailed experimental metadata.
Reference
- Ma, L.; Ma, J.; Xu, K. Effect of Spaceflight on the Circadian Rhythm, Lifespan and Gene Expression of Drosophila melanogaster. PLoS One 2015, 10, doi:10.1371/journal.pone.0121600.
Point 3: In Figure 1A you showed a gene expression signature which is likely the basis of this analysis. With exception of Arntl, none of these genes are described in the results section and / or in the discussion. However, other genes are found there (Cry, Per2, ….). Why this discrepancy?
Response 3: For the tissue-wide analysis in Figure 1, a comparative analysis between FLT and GC (regardless of tissue type) conditions was performed to examine how spaceflight affects prevalent molecular mechanisms across tissues. Figure 2 shows a more detailed comparative analysis between FLT and GC in each of the eight tissues. Some clock genes were not observed to change expression in some tissues in these eight individual tissue comparative analyses. Therefore, there is a discrepancy between the results of the single tissue-wide and eight individual tissue comparative analyses. However, enrichment analyses show that the terms associated with circadian rhythms are consistent between the tissue-wide and individual comparative analyses. Therefore, clock genes are likely to be altered by spaceflight throughout the tissues.
Some core clock genes we have added to the discussion according to your suggestion below.
“Arntl, Clock, Per2, Per3, Cry1, and Cry2 were enriched across some tissues in MODE1 of the PPI network analysis (Figure 3B). In addition, Arntl, Clock, and Per3 were significantly changed in the tissue-wide analysis (Figure 1A). Since TFs of these genes form core clock components in circadian rhythm mechanisms [31], these clock genes could play a role as key regulators associated with underlying mechanisms of physiology and metabolism under spaceflight [18,52]. For example, previous studies showed that clock genes have demonstrated a role of circadian rhythm in muscle atrophy and bone remodeling [53–55]. In addition, these clock genes have been implicated in NAFLD [56], the symptoms of which have been observed in mice during spaceflight [10]. Furthermore, Arntl plays a role in the circadian regulation of acute glucocorticoid secretion in the adrenal glands in response to stress [57]. Mice models carrying a conditional allele for Arntl showed several disorders, such as increased urine volume, changes in the circadian rhythm of urinary sodium excretion, increased glomerular filtration rate, and significantly reduced plasma aldosterone levels [58]. Clock null mice showed abnormal circadian rhythmicity of plasma aldosterone levels and changes in circadian gene expression patterns in the kidney [59]. Glucocorticoids could shift the phase of circadian oscillations of Per1 and Per2 expressions in peripheral tissues [60]. Cry genes are associated with changes in the transcriptional response to glucocorticoids in mouse embryonic fibroblasts [61]. In addition, Cry1 and Cry2 null mice indicated salt-sensitive hypertension via abnormally high synthesis of the mineralocorticoid aldosterone by the adrenal gland [62]. Given that circulating stress hormones are increased and circadian rhythms are altered in astronauts and mice [34,35,63], spaceflight could contribute to circadian rhythm disruption and asynchrony between peripheral tissues. Therefore, these results suggest that these clock genes could modulate physiology and metabolism mechanisms as key regulators during spaceflight environmental changes. However, the role of clock genes is unknown in peripheral tissues under spaceflight. In a simulated microgravity study, for example, Arntl disrupted diurnal oscillation in rat cerebrovascular contractility by changing circadian regulation of the miR-103/CaV 1.2 signal pathway [64]. Oscillations of Arntl were amplified under simulated microgravity in human keratinocytes [65]. Future studies could determine clock gene roles in peripheral tissues under spaceflight, such as by using clock genes null mice under spaceflight [66].”

Round 2
Reviewer 2 Report
In its revised form, the manuscript by Fujita et al. is suitable for publication. All suggestions were implemented accurately.